Graph convolutional network and self-attentive for sequential recommendation

http://orcid.org/0009-0004-2484-1004 Guo Kaifeng 832103126@fzu.edu.cn
Zeng Guolei
Fuzhou University , Fuzhou, Fujian , China
Kong Xiangjie
Electronic publication date: 2023 Dec 1
Publication date: 2023
Volume: 9
Electronic Location ID: e1701
Received 2023 Aug 16; Accepted 2023 Oct 25
Copyright: © 2023 Guo and Zeng
Copyright year: 2023
Copyright holder: Guo and Zeng
License: This is an open access article distributed under the terms of the Creative Commons Attribution License, which permits unrestricted use, distribution, reproduction and adaptation in any medium and for any purpose provided that it is properly attributed. For attribution, the original author(s), title, publication source (PeerJ Computer Science) and either DOI or URL of the article must be cited.
License URL: https://creativecommons.org/licenses/by/4.0/

Keywords: Sequential recommendation, Contrastive learning, Graph convolutional network, Deep learning

Funding: Research Training Program of Fuzhou University 29373 This work was supported by the research training program of Fuzhou University (No. 29373). The funders had no role in study design, data collection and analysis, decision to publish, or preparation of the manuscript.

==============================
Sequential recommender systems (SRS) aim to provide personalized recommendations to users in the context of large-scale datasets and complex user behavior sequences. However, the effectiveness of most existing embedding techniques in capturing the intricate relationships between items remains suboptimal, with a significant concentration of item embedding vectors that hinder the improvement of final prediction performance. Nevertheless, our study reveals that the distribution of item embeddings can be effectively dispersed through graph interaction networks and contrastive learning. In this article, we propose a graph convolutional neural network to capture the complex relationships between users and items, leveraging the learned embedding vectors of nodes to represent items. Additionally, we employ a self-attentive sequential model to predict outcomes based on the item embedding sequences of individual users. Furthermore, we incorporate instance-wise contrastive learning (ICL) and prototype contrastive learning (PCL) during the training process to enhance the effectiveness of representation learning. Broad comparative experiments and ablation studies were conducted across four distinct datasets. The experimental outcomes clearly demonstrate the superior performance of our proposed GSASRec model.

Introduction

Personalized recommendation has become a dominant and widely adopted approach in various real-world applications, empowering users with tailored item suggestions that cater to their individual interests (Cheng et al., 2016; Fayyaz et al., 2020). The core task of a recommender system revolves around predictive modeling, which aims to predict the likelihood of user-item interactions, encompassing various forms of engagement like clicks, ratings, and purchases, among others. This predictive capability serves as the foundation of effective recommendation systems, enabling them to provide users with relevant and appealing item recommendations, thereby enhancing user satisfaction and engagement. Therefore, accurately capturing user preferences is a critical aspect (Peng, Sugiyama & Mine, 2022).

Collaborative filtering (CF) has emerged as a potent solution for recommendation systems. It relies on historical user-item interactions, such as purchases or clicks, with the assumption that users with similar behavior are likely to exhibit similar preferences for items (Cheng et al., 2018; He et al., 2017). One of its key advantages is that it does not rely on explicit feature engineering or content analysis, allowing it to discover hidden patterns and relationships between users and items solely based on user interactions. This approach makes collaborative filtering a powerful method to make personalized recommendations in various domains. Extensive research on CF-based recommenders has been conducted, leading to remarkable achievements in this field (Koren, 2008; He et al., 2018; Wang et al., 2019; He et al., 2020). However, collaborative filtering methods cannot directly consider the temporal relationships of user behaviors, which means they may not capture the evolving patterns of user behavior over time, thus performing suboptimally in handling recommendation problems involving temporal dependencies. On the other hand, Sequential recommendation (SR) is a branch of recommendation systems that focuses on providing personalized recommendations by considering the temporal order of user behavior sequences. User interests and preferences are known to evolve and change gradually. To handle the temporal dependency issues in SR, researchers have developed specialized models such as BERT4Rec (Sun et al., 2019) and SASRec (Kang & McAuley, 2018). These models organize users’ actions, such as browsing, purchasing, adding to cart, and other interactions, in chronological order and employ attention mechanisms or positional encoding to gain a better understanding of how user interests evolve over time. Numerous sequential recommendation (Xie et al., 2020; Chen et al., 2022; Zhou et al., 2020; Liu et al., 2021a; Li et al., 2023) studies delve into exploring more effective ways of representing embedded representations of sequential items. One such approach is contrastive learning (Chen et al., 2020) where a positive sample sequence is obtained through sequence augmentation methods, while other sequences serve as negative samples. By encouraging the model to increase the similarity between the encodings of positive sample sequences and decrease the similarity with negative sample sequences, the model’s representational capacity is enhanced. Consequently, the model becomes better equipped to differentiate between the long-term and short-term interests and intentions of distinct users. Through this approach, the model gains a more comprehensive understanding of the intricate patterns embedded in users’ sequential behaviors, thus yielding more accurate and personalized recommendations.

However, sequential recommendation models often face challenges in directly learning the similarities between users and items, as well as item-item and user-user relationships. In contrast, collaborative filtering methods, such as multi-layer graph convolutions on user-item interaction graphs, can effectively unearth the underlying connections between items and users. For instance, users with similar behavior sequences are likely to have similar embedded representations, leading to higher similarity scores. Consequently, if two users have similar embedded representations for certain items, their overall item representations should also exhibit a higher degree of similarity.

Therefore, by combining collaborative filtering with sequential recommendation, we can address these issues. In this regard, we propose a method that utilizes user interaction graph convolutions to extract item embeddings and then employs a sequential recommendation model to predict the user’s next actions. To further enhance the model’s effectiveness, we incorporate instance contrastive learning and prototype contrastive learning to improve its representational capacity.

In summary, this article makes several contributions are: We propose that combining interactive graphs and attention-based sequence models can complement each other’s limitations. We have empirically demonstrated that the fusion of these two techniques can indeed effectively enhance model performance.

During the training phase, we employ a multi-task learning approach by integrating instance-wise contrastive learning and prototype contrastive learning. We have verified that the combination of these two contrasting learning methods can further improve model effectiveness.

Extensive experiments are carried out on four widely-used public datasets, showcasing the consistent superiority of our proposed approach over various competitive baselines. Additionally, we conducted multiple sets of ablation experiments to validate the effectiveness of each module.

Related work

Collaborative filtering

Collaborative filtering (CF) is a popular approach in recommendation systems that involves learning latent features, or embeddings, to represent users and items. The prediction is then performed based on these embedding vectors. Matrix factorization is one of the early CF models, where users’ interaction history is not explicitly considered, and only the user ID is projected to the embedding. However, subsequent research has shown that incorporating user interaction history can improve the quality of embeddings and prediction performance.

An example of this is the utilization of user interaction history in predicting numerical ratings, as demonstrated by SVD++ (Koren, 2008). Additionally, Neural Attentive Item Similarity (NAIS) assigns varying degrees of importance to items present in the interaction history, leading to more accurate item ranking predictions (He et al., 2018). The key to these enhancements lies in leveraging the subgraph structure of a user’s interaction history, particularly considering their one-hop neighbors, which effectively enhances the process of embedding learning.

To further leverage the subgraph structure, Wang et al. (2019) propose NGCF, a state-of-the-art CF model inspired by graph convolution network (GCN) (Wu et al., 2019). NGCF adopts the propagation rule of GCN, which involves feature transformation, neighborhood aggregation, and nonlinear activation, to refine embeddings. While NGCF has shown promising results, it inherits many operations from GCN without justifying their relevance to the CF task. This design choice introduces unnecessary complexity, particularly when applied to user-item interaction graphs, where each node has only a one-hot ID without rich attribute information. LightGCN (He et al., 2020) introduces a novel approach that propagates user and item embeddings linearly onto the user-item interaction graph, leveraging the weighted summation of embeddings learned across all layers as the ultimate embedding. This method exhibits significant performance improvements over NGCF, as evidenced by our experimental results.

Sequential recommendation

Sequential recommendation has garnered significant research attention in recent years, aiming to accurately capture users’ dynamic interests by modeling their past behavior sequences. Early approaches in this field focused on utilizing Markov chains to model item-to-item transaction patterns. For instance, FPMC combined Markov chains with matrix factorization techniques to integrate sequential patterns and users’ general interests (Rendle, Freudenthaler & Schmidt-Thieme, 2010).

In light of the rise of deep learning, a multitude of deep sequential recommendation models have emerged, harnessing neural networks to capture both long-term and short-term preferences from behavioral sequences. Recurrent neural networks (RNNs) gained prominence due to their ability to encode sequential dependencies. For example, GRU4Rec employed gated recurrent units (GRUs) to model user interests (Hidasi et al., 2015). Another avenue of research delved into the use of convolutional neural networks (CNNs) for sequential recommendation (Yan et al., 2019).

The success of attention mechanisms in natural language processing tasks has motivated its adoption in sequential recommendation. Attention-based models have shown promise in capturing complex dependencies in behavior sequences. SASRec introduced the use of unidirectional attention mechanisms to assign adaptive weights to interacted items (Kang & McAuley, 2018). BERT4Rec improved upon this approach by employing bidirectional attention mechanisms with a Cloze task (Sun et al., 2019). LSAN proposed a light-weight approach with a temporal context-aware embedding and a twin-attention network (Li et al., 2021). ASReP addressed data sparsity by leveraging a attention mechanism on revised user behavior sequences (Liu et al., 2021b). DuoRec (Qiu et al., 2022) introduces innovative techniques to improve semantic preservation and address the representation degeneration problem in recommendation systems.

Contrastive learning for recommendation

Contrastive learning (CL) has garnered significant attention in various research domains such as computer vision, natural language processing, and recommender systems. In the context of recommender systems, the focus of contrastive learning lies in optimizing mutual information between positively transformed data samples while simultaneously enhancing the discriminability of negative samples. Traditional recommender systems often rely on large amounts of labeled user behavioral data, which are often difficult to obtain and may result in subpar recommendations for new users and rare items. In contrast, contrastive learning, with its label-free self-supervised learning approach, exhibits remarkable advantages in recommender systems.

Early works in contrastive learning for recommendation focused on utilizing deep neural networks (DNNs) to enhance collaborative filtering-based recommendation leveraging item attributes (Yao et al., 2020). These models utilized a two-tower architecture to compare positive and negative samples and learn effective item representations. Another line of research employed contrastive learning within graph neural networks (GCNs) to improve collaborative filtering methods using only item IDs as features (Wu et al., 2020).

In the domain of sequential recommendation, contrastive self-supervised learning (SSL) has been utilized to capture associations among items, subsequences, and characteristics found in user behavior sequences (Zhou et al., 2020). These models adopt an end-to-end training approach, incorporating contrastive SSL throughout the entire training phase. Nonetheless, this unified training methodology facilitates information sharing between the SSL and next-item prediction tasks, eliminating the need for separate fine-tuning and pre-training stages, potentially constraining overall performance enhancement. To overcome this limitation, recent studies have proposed multi-task training frameworks incorporating a contrastive objective to improve user representations (Xie et al., 2020; Liu et al., 2021a). Furthermore, a novel approach named ICLRec, presented by Chen et al. (2022), introduces clustering techniques to extract users’ intent distributions from their behavior sequences. By leveraging clustering, ICLRec identifies distinct patterns of user intent embedded within the data.

Preliminaries

Problem settings

Let V and U represent the sets of items and users, respectively. We denote a user u∈U interaction sequence as Su={v1,v2,....,vT}, where T is the total number of items in the sequence, and the items are ordered chronologically. Each item vi∈Su is associated with an order index i=1,2,...,T, indicating its position in the sequence. Our objective is to create a prioritized list of the top K items that user u is highly likely to visit in the subsequent time step T + 1.

Proposed model

In this section, we will introduce our proposed graph convolution and self-attention model, named GSASRec. GSASRec is primarily composed of interaction graph convolution (IGC) layers and self-attention layers. We will proceed to describe each layer of the model in the order of forward propagation, along with the contrastive learning methods utilized in the model.

Embedding layer

We expound on the representation of a user, denoted as u, and an item, denoted as i, through their respective embedding vectors, eu∈Rd (for user u) and ei∈Rd (for item i), where d signifies the embedding dimension. The described process can be the creation of a parameter matrix, which operates akin to an embedding look-up table:

Eu=[eu1,eu2,...,eut]

Ei=[ei1,ei2,...,eim]

where t represents the total number of users, while m corresponds to the total number of items. For the input sequence Su={v1,v2,...,vn}, data augmentation techniques such as masking, cropping, noising, and reordering are applied to obtain two augmented sequence Su′={v1′,v2′,...,vn′} and Su′′={v1′′,v2′′,...,vn′′}. Then, based on the Ei table, we can acquire their embedding ESu={ev1,ev2,...,evn}∈Rn×d, ESu′={ev1′,ev2′,...,evn′}∈Rn×d and ESu′′={ev1′′,ev2′′,...,evn′′}∈Rn×d.

Interaction graph convolution layer

LightGCN (He et al., 2020) incorporates graph convolution neural networks into collaborative filtering, taking into account the latent relationships between users and items, as well as between items themselves. However, during prediction, it does not consider the temporal order of item sequences. Therefore, in this work, we leverage graph convolution neural networks to extract latent embedding information, with a focus on capturing the sequential characteristics of items, as illustrated in Fig. 1.

Figure 1 The overview of interaction graph convolution layer.

Based on the training data, we construct the user-item interaction matrix R∈Rt×m and the item-user interaction matrix RT∈Rm×t. With these matrices in place, we define the graph convolution network as follows:

eu(k+1)=∑i∈Nu1|Nu||Ni|ei(k)

ei(k+1)=∑u∈Ni1|Ni||Nu|eu(k)

where ei(k+1) represents the updated representation of node i in the k+1st iteration. The sum is taken over all the neighboring nodes u of node i denoted by Ni. The term 1|Ni||Nu| is a normalization factor that accounts for the degree of nodes i and u, and eu(k) is the representation of node u in the k-th iteration. When k=0, we initialize ei(0)=ei∈Ei and eu(0)=eu∈Eu. This update rule is used in graph convolution networks to aggregate neighboring node features and update the representation of each node in the graph.

As items undergo multiple graph convolutions, and the sequence model focuses solely on item sequences for recommendations, we extract only the item embedding representations for the subsequent layers. We aggregate the outputs of various convolution layers to obtain the graph embedding representation for item i.

ei(k)=1K∑k=0Kei(k)

where K represents the number of graph convolution layers utilized in the model.

Self-attention layer

To represent the temporal order within a sequence, we employ positional embedding. Assuming the positional embedding is represented as P∈ℝn×d, we add it to the embedding of the behavioral sequence:

Ep^=[ev1(k)+P1ev2(k)+P2⋯evn(k)+Pn]Ep^′=[ev1′(k)+P1ev2′(k)+P2⋯evn′(k)+Pn]Ep^′′=[ev1′′(k)+P1ev2′′(k)+P2⋯evn′′(k)+Pn]

we incorporate self-attention mechanism and feed-forward network layers:

EA=Attention(Ep^WQ,Ep^WK,Ep^WV)

F=ReLU(EAW(1)+b(1))W(2)+b(2)

where the matrices WQ,WK,WV∈Rd×d and the matrices Q,K,V∈Rn×d. W(1) and W(2)∈Rd×d serve as parameter matrices, while b(1) and b(2)∈Rd represent bias vectors. The attention mechanism is expressed as follows:

Attention(Q,K,V)=softmax (QKTd)V

Similarly, from Ep^′ and Ep^′′, we can obtain F′ and F′′.

Recommendation learning

For the output sequence F={f1,f2,...,fn} of the feed-forward network (FFN), we can compute the binary cross-entropy loss at each step of the recommendation model:

LRec=−∑t∈[1,2,…,n−1]log(σ(ft⋅evt+1))−log⁡(σ(fn⋅evy^))−∑t∈[1,2,…,n]∑j∉Sulog(1−σ(ft⋅evj))

where fi represents the output of the i-th FFN of the model. y^ represents the index of the target item at position n+1 in the sequence within Ei. σ denotes the sigmoid function. evy^ signifies the embedding representation of the training label.

Instance-wise contrastive learning

For a training batch B={F1,F2,...,Fb,F1′,F2′,…Fb′,F1′′,F2′′,…,Fb′′}, where b is the number of original sequences, and 3⋅b is the total number of sequences in one batch, comprising the original sequences and their two augmented sequences, we aim to maximize the similarity between Fi and its corresponding augmented sequences Fi′ and Fi′′, as well as the similarity between the two augmented sequences themselves. Additionally, we seek to minimize the similarity between Fi, Fi′, Fi′′, and the other sequences in the batch, thereby achieving contrastive learning. Hence, we can compute the InfoNCE loss for the batch B:

LCL(Fi,Fi′)=−loge(sim(Fi,Fi′))/τ∑j=1,j≠ib[esim(Fi,Fj)/τ+esim(Fi,Fj′)/τ+e(sim(Fi,Fj′′))/τ]

LICL=∑i=1b[LCL(Fi,Fi′)+LCL(Fi′,Fi′′)+LCL(Fi′′,Fi)]

where sim(⋅) represents the tensor similarity function, which is used to calculate the similarity between tensors.

Prototype contrastive learning

Prototype contrastive learning aims to learn feature representations by comparing the similarity between samples and prototypes. This process makes the feature representations of similar samples closer while pushing those of dissimilar samples further apart, resulting in the formation of distinct clusters. Typically, this learning is conducted after multiple rounds of training, specifically when the instance contrastive learning loss approaches relative stability. We interpret the embedding encoding of a user’s entire sequence as the representation of their long-term interest. Generally, users with similar behavioral sequences exhibit close long-term interest embeddings. Hence, adopting prototype contrastive learning can bring the embedding encodings of similar behavioral sequences closer, placing them within the same category. This approach is advantageous for recommendation systems as it facilitates recommending similar items to users with shared interests.

We apply k-means clustering M times to the embedding representations of all user sequences in the data. For each iteration m (1≤m≤M), we randomly select several points as the initial cluster centroids, denoted as C={c1m,c2m,...,c|C|m}. After several iterations of clustering in the m-th run, we fix the cluster centroids. Subsequently, we define the function:

g(f¯i)=arg⁡minjd(f¯i,cjm)

where d(f¯i,cjm)=∑k=1d(f¯i,k−cj,km)2, f¯i=1|Fi|∑fi∈Fifi, and the function g(⋅) assists in identifying the nearest cluster centroid cj for each averaged embedding f¯i calculated as the mean of all embeddings fi within the set Fi. We leverage pre-iterated cluster centers for contrastive learning and compute the loss function as follows:

LPCLm(f¯i,cg(f¯i)m)=−log⁡e(f¯i⋅cg(f¯i)m)∑j=0,j≠g(f¯i)|C|e(f¯i⋅cjm).

LPCL(f¯i)=1M∑m=0MLPCLm(f¯i,cg(f¯i)m)

Multi-task learning

To enhance model performance, data efficiency, and generalization capability, and to address challenges such as data scarcity and overfitting, we adopt a multitask learning approach, as shown in Fig. 2, to integrate recommendation, instance contrastive learning, and prototype contrastive learning tasks. Specifically, we jointly optimize the loss functions of these tasks:

Figure 2 The overview of GSASRec in the training stage.

We assume that the input sequence of examples goes through data augmentation techniques, such as introducing noise, to generate two positive sample sequences (only one is shown in the figure). In this process, we randomly replace i9 and i6 with i7 and i2, respectively. Subsequently, the encoded sequences undergo multitask learning, involving instance contrastive learning and prototype contrastive learning.

L=LRec+λ⋅LICL+β⋅LPCL

where λ and β are adjustable parameters used to balance the importance of the losses.

Experiments

In this section, an extensive assessment is conducted to evaluate the recommendation efficacy of our GSASRec model, designed for sequential recommendation tasks. Our evaluation entails a comprehensive analysis that includes a comparative study between GSASRec and previous sequential recommenders. Subsequently, we delve into a thorough investigation to explore the influence of crucial components and hyperparameters integrated within GSASRec’s architecture. This systematic examination aims to shed light on the model’s strengths and potential areas for further enhancement, contributing to the advancement of sequential recommendation techniques powered by deep learning methodologies.

Experimental setting

Datasets

In our investigation, we embark on a series of experiments encompassing four widely adopted benchmark datasets. These datasets have their statistical attributes meticulously summarized and displayed in Table 1. Incorporated within McAuley et al. (2015), the Amazon review dataset has been thoughtfully partitioned into three distinct subcategories, namely Sports, Beauty and Toys. Concurrently, Yelp emerges as a prominent dataset tailored for the specific task of business recommendation. Following the methodology presented in reference (Xie et al., 2020), we adopt a similar approach to preprocess the dataset, eliminating users with fewer than five interactions.

Table 1 Statistics of experimental datasets.

Dataset	#Users	#Items	#Interactions	Density (%)	
Sports	35,598	18,357	296,337	0.05	
Beauty	22,363	12,101	198,502	0.07	
Toys	19,412	11,924	167,597	0.07	
Yelp	22,845	16,552	243,703	0.06	
Note:

Density (%) = #Interactions#Users×#Items.

Evaluation metrics

To evaluate the performance of our approach, we utilize two widely recognized Top-K metrics (NDCG@K and HR@K) as proposed by a previous work (Krichene & Rendle, 2020). The formula for NDCG@K is as follows:

NDCG@K=DCG@KIDCG@K

where DCG@K=∑i=1Krelilog2(i+1) and reli is the relevance score of the item at position i in the ranked list. IDCG@K is the maximum possible DCG@K achievable for a perfect ranking. It is calculated by sorting the items by their true relevance scores in descending order and then calculating DCG@K for this ideal ranking.

HR@K is a binary evaluation metric, commonly used for the performance evaluation of recommendation systems. The formula for HR@K is as follows:

HR@K=NumberofrelevantitemsinrecommendationsK

where the number of relevant items in recommendations is the number of items related to user interests in the first K recommended results.

Overall, HR@K measures the percentage of recommended items that contain at least one ground truth item within the top K positions. On the other hand, NDCG@K assesses the ranking quality by giving higher scores to hits at higher-ranked positions. These metrics provide a quantitative measure of how effective each model is at recommending relevant items within the top K positions. By comparing NDCG@K or HR@K scores, we can determine which model is better at surfacing relevant content to users. Higher scores indicate more effective recommendations. To ensure consistency, we set the value of K to 5 and 10 for both metrics.

Baseline methods

We compare GSASRec with the following baseline methods: BPR-MF (Rendle et al., 2012) proposed a generic learning algorithm based on stochastic gradient descent with bootstrap sampling.

Caser (Tang & Wang, 2018) proposed a convolutional sequence embedding recommendation model, which effectively captures both general preferences and sequential patterns in recommendation tasks.

GRU4Rec (Hidasi et al., 2015) proposed a novel session-based recommendation model based on GRUs, which effectively captures temporal dependencies in user behavior sequences.

SASRec (Kang & McAuley, 2018) utilized self-attention mechanism for sequential recommendation.

BERT4Rec (Sun et al., 2019) adopted BERT as the sequential recommendation model.

S3Rec (Zhou et al., 2020) adopted a self-supervised learning approach, where items in the user behavior sequence are masked, and the masked sequence is used to predict the masked items.

CL4SRec (Xie et al., 2020) proposes the use of data augmentation in contrastive learning to enhance the effectiveness of recommendation systems.

ICLRec (Chen et al., 2022) leveraged clustering to learn user intent and validated the rationality of this approach.

Implementation

We employ various critical hyperparameters. Specifically, we configure the embedding size to 64, establish the maximum sequence length at 50, define a batch size of 256, and specify 300 epochs for training. When it comes to the contrastive learning loss during prototype computation, our learning process kicks off from epoch 160, with a learning rate set at 0.001. Our model architecture comprises three graph convolutional layers, each of which incorporates two self-attention blocks with two attention heads. We set λ to 0.9 and β to 0.1. Additionally, we iterate through the clustering procedure M times, with M being defined as 3. Furthermore, we harness the PyTorch framework, and our GPU is equipped with an NVIDIA GeForce RTX 3070, supported by a substantial 64 GB of computer RAM.

Overall performance

Through the analysis of Table 2, we can observe the results obtained by various methods on different datasets. we observe that incorporating sequential patterns in user behavior sequences enhances the performance of sequential models like SASRec and Caser, surpassing the non-sequential approach BPR-MF. This highlights the significance of mining sequential patterns, with GRU4Rec also exhibiting improved results over BPR-MF in the deep learning era. Furthermore, Caser, leveraging a convolutional module to stack sequential tokens as a matrix, performs on par with GRU4Rec. Moreover, SASRec stands out as the pioneer in utilizing uni-directional attention for sequence encoding, demonstrating its superiority over previous deep learning-based models by significantly improving performance. With the rise of contrastive learning techniques in recommendation systems, BERT4Rec, S3-Rec, and CL4SRec have all leveraged contrastive learning to enhance model performance, surpassing pure sequential recommendation models. However, the two-stage training strategy employed in S3-Rec obstructs information sharing between tasks, resulting in suboptimal outcomes. On the contrary, CL4SRec consistently outperforms other baselines, showcasing the efficacy of contrastive self-supervised learning in enriching sequence representations at an individual user level. The additional objective employed by CL4SRec, entailing two distinct views of the same sequence, significantly contributes to its superior performance. Subsequently, the emergence of ICLRec method combines the advantages of previous approaches and introduces user intent extraction techniques, which also rely on contrastive learning methods, resulting in significant improvements. Finally, our proposed GSASRec model achieves even greater improvements compared to ICLRec. In contrast, we enhance the model’s representational capacity by leveraging graph convolutional techniques on the user-item interaction graph. Moreover, we perform multiple prototype clustering to mitigate noise interference and introduce a data augmentation method for instance-based contrastive learning.

Table 2 Overall performance.

Dataset	Metric	BPR	GRU4Rec	Caser	SASRec	BERT4Rec	S3Rec	CL4Rec	ICLRec	GSASRec	Improve	
Sports	HR@5	0.0101	0.0136	0.0140	0.0219	0.0177	0.0158	0.0229	0.0282	0.0306 ± 0.0012	8.51%	
HR@10	0.0194	0.0278	0.0231	0.0336	0.0326	0.0265	0.0373	0.0431	0.0462 ± 0.0008	7.19%	
NDCG@5	0.0048	0.0096	0.0086	0.0128	0.0105	0.0098	0.0131	0.0182	0.0209 ± 0.0006	14.83%	
NDCG@10	0.0063	0.0136	0.0126	0.0169	0.0155	0.0135	0.0185	0.0230	0.0258 ± 0.0005	12.17%	
Beauty	HR@5	0.0134	0.0165	0.0258	0.0367	0.0194	0.0327	0.0402	0.0493	0.0518 ± 0.0011	5.07%	
HR@10	0.0301	0.0365	0.0421	0.0627	0.0401	0.0594	0.0686	0.0736	0.0788 ± 0.0013	7.06%	
NDCG@5	0.0045	0.0087	0.0131	0.0236	0.0189	0.0176	0.0231	0.0324	0.0344 ± 0.0007	6.17%	
NDCG@10	0.0058	0.0143	0.0256	0.0281	0.0254	0.0269	0.0318	0.0401	0.0426 ± 0.0010	6.23%	
Yelp	HR@5	0.0131	0.0154	0.0156	0.0161	0.0186	0.0175	0.0218	0.0245	0.0257 ± 0.0004	4.90%	
HR@10	0.0246	0.0265	0.0254	0.0265	0.0292	0.0283	0.0354	0.0408	0.0429 ± 0.0012	4.91%	
NDCG@5	0.0760	0.1070	0.0097	0.0101	0.0118	0.0115	0.0131	0.0153	0.0161 ± 0.0005	5.22%	
NDCG@10	0.0119	0.0136	0.0129	0.0135	0.0173	0.0162	0.0188	0.0207	0.0216 ± 0.0008	4.34%	
Toys	HR@5	0.0120	0.0098	0.0164	0.0467	0.0277	0.0144	0.0536	0.0590	0.0621 ± 0.0018	5.25%	
HR@10	0.0206	0.0177	0.0274	0.0655	0.0449	0.0553	0.0816	0.0834	0.0863 ± 0.0035	3.47%	
NDCG@5	0.0081	0.0061	0.0109	0.0310	0.0177	0.0131	0.0369	0.0406	0.0421 ± 0.0013	3.69%	
NDCG@10	0.0113	0.0173	0.0274	0.0649	0.0198	0.0371	0.0434	0.0481	0.0502 ± 0.0015	4.36%	
Notes:

Bold indicates the best result among all methods, while underlining represents the highest result among previous methods.

Improve(%)=Ourmodelscore−highestresultamongpreviousmethodshighestresultamongpreviousmethods.

Figure 3 presents the model’s performance at each epoch. It is important to highlight that we introduced the prototype contrastive learning loss at epoch 160, as depicted in Fig. 3E. This led to a noticeable increase in the computed loss values, resulting in distinctive fluctuations and an overall upward trend in the curves, particularly evident in the Toys (Fig. 3C) and Beauty (Fig. 3B) datasets.

Figure 3 The training curves of GSASRec, which are evaluated through training loss, and testing HR@k and NDCG@k per epoch on the Sports, Beauty, Toys, and Yelp datasets.

Ablation study

Impact of parameters λ and β

As shown in Fig. 4, to evaluate the impact of loss function weights on the model’s performance, we conducted experiments with multiple sets of λ and β values and assessed the model’s NDCG@10 performance on four different datasets. The results indicate that the model performs best when λ is set to 0.9 and β to 0.1. However, when β is set to 0 or greater than 0.1, the model’s performance deteriorates. We attribute this to the introduction of the prototype contrast loss, which results in the prototype contrast loss value becoming much larger than the instance contrast loss value after a certain number of epochs. Consequently, the model overly emphasizes the prototype contrast task and does not continue to optimize the sequence recommendation task and the instance contrast task. Therefore, it is necessary to reasonably reduce the weight of the prototype contrast task.

Figure 4 The performance of ablation experiments on the parameters λ and β.

Impact of model components on recommendation performance

To validate the effectiveness of various model architectures and methods thoroughly, we conducted comprehensive ablation experiments on four diverse datasets, leveraging the widely accepted NDCG@10 metric for evaluation. The conducted ablation experiments involved systematically removing specific functionalities from our proposed model, GSASRec, in order to gauge their individual contributions to the overall performance.

In Fig. 5, we present the insightful results obtained from these ablation experiments. Each abbreviation in the figure represents a specific functionality removed from the GSASRec model. ‘w/o’ stands for ‘without,’ indicating the absence of the corresponding functionality. Specifically, ‘ICL’ represents Instance-wise contrastive learning, ‘PCL’ refers to prototype contrastive learning, ‘IGCL’ signifies the interaction graph convolution layer and ‘SAL’ denotes the self-attention layer, when the self-attention layer is removed, we employ the embeddings of individual items obtained from the graph convolution layer (GCL) as the objects for contrastive learning. Additionally, we employed ‘ICLRec’ as the baseline model, representing the best-performing model from the instance-wise contrastive learning methods. The outcomes of the ablation experiments are highly informative. It is evident that instance-wise contrastive learning has a substantial and positive impact on the model’s overall performance, indicating its crucial role in enhancing recommendation accuracy. Following closely is the self-attention layer, which also demonstrates its significance in contributing to improved recommendation results. Moreover, the results highlight the importance of prototype contrastive learning, particularly for the Toys dataset, where it exhibits a noteworthy influence on enhancing recommendation performance. This observation emphasizes the versatility of our proposed model across different datasets and the potential of prototype contrastive learning in addressing specific domain challenges. Furthermore, the interaction graph convolution layer stands out as a significant component in our model, consistently leading to substantial performance improvements across all the evaluated datasets. This finding underlines the efficacy of incorporating graph-based interactions to capture complex relationships between users and items, reinforcing the importance of leveraging graph-based learning methods in recommendation systems.

Figure 5 The performance of ablation experiments on the four datasets.

Impact of layer combination

In our model, we aggregate the outputs of k convolutional layers to obtain the embedding representation. The choice of different k values significantly influences the effectiveness of the model’s embedding representation. Figure 6 illustrates the effects of varying convolutional layer depths on four distinct datasets. The term ‘Number of Layer’ corresponds to the upper limit of k values in the model. The curve trends in the figure are generally consistent, with the model’s performance peaking at the third layer in most cases. However, for the Beauty dataset (Fig. 6B), the performance of the four-layer graph convolutional network slightly outperforms that of the third layer. Consequently, we can infer that the suboptimal performance for k values below three may stem from the model’s inability to fully capture the complex relationships and features present in the graph data. Shallow graph convolutional networks might be limited in their ability to propagate information among local neighbor nodes, making it difficult to capture global structures and longer-range dependencies. As the number of convolutional layers increases, the model progressively expands its receptive field, utilizing more extensive graph structure information for feature propagation and learning. Nevertheless, excessively deep networks may encounter issues of vanishing or exploding gradients, leading to a plateau in model performance beyond 3 or 4 layers.

Figure 6 The results of different graph convolution layer settings in the four datasets.

Conclusions

In this article, we present a novel sequence recommendation model that integrates interactive graph convolutional networks (GCNs) and employs various contrastive learning techniques to enhance its performance. Specifically, we leverage multiple layers of graph convolution to capture latent relationships between items in the user-item interaction graph. The outputs of these graph convolution layers are aggregated to obtain item embedding representations. Furthermore, we incorporate attention mechanisms and position embedding encoding into the sequence model, combining the advantages of interactive graph convolutions with sequence recommendation models. To further improve the model’s representation capabilities, we employ instance contrastive learning and prototype contrastive learning techniques. The introduction of these contrastive learning techniques enables our model to better capture the underlying structures and patterns in the data, leading to improved recommendation performance. We have conducted extensive comparative experiments and ablation studies to demonstrate the superiority of our proposed method.

Supplemental Information

Supplemental Information 1 Source code.

To ensure a fair comparison of models, the majority of the code in the files datasets.py, utils.py, modules.py, main.py, and data_augmentation.py is adapted from the code provided by S3-Rec and ICL-Rec. However, we have implemented the code for models.py and trainers.py according to the approach we have proposed.

Click here for additional data file.

Additional Information and Declarations

Competing Interests

Author Contributions

Data Availability

The authors declare that they have no competing interests.

Kaifeng Guo conceived and designed the experiments, performed the experiments, analyzed the data, performed the computation work, prepared figures and/or tables, authored or reviewed drafts of the article, and approved the final draft.

Guolei Zeng performed the experiments, analyzed the data, prepared figures and/or tables, and approved the final draft.

The following information was supplied regarding data availability:

The code is available in the Supplemental File.

The data is available at Recommender Systems and Personalization Datasets (https://cseweb.ucsd.edu/~jmcauley/datasets.html#amazon_reviews) and Yelp Open Dataset (https://www.yelp.com/dataset).

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
