# Peer review of "Graph convolutional network and self-attentive for sequential recommendation"

_PeerJ Computer Science, doi:10.7717/peerj-cs.1701_

## Round 0.1 · original submission · Major Revisions

The structure of the work is clear. However, the reviewers also pointed out that some details should be added and the writing should be double-checked and improved. Please revise the article accordingly.

Reviewer 1 ·

Basic reporting

The general structure of this paper is clear. Some issues are expected to be checked.
(1) The title of table 2 is confused, seems that the title does not fit to the content in Table 2.
(2) In Table 2, please clarify how does the value in the 'Improve' column calculated, which baseline is compared to GSARec?
(3) All the upper quotation marks are displayed as the lower quotation marks.
(4) some punctuations are missed between line 111 to 117. Other typos and formats should be carefully checked.

Experimental design

(1) Section 1.7 discusses λ and β in the formulas. Please provide additional experimental details to show how these parameters are determined.
(2) Section 1.8.4 should provide information about the hardware and software environment used for conducting the experiments, including any specific configurations or settings.

Validity of the findings

While the article introduces intriguing viewpoints and research outcomes, there is still potential for enhancement in terms of its contributions. Please emphasize the uniqueness and significance of this research more explicitly.

Cite this review as

Reviewer 2 ·

Basic reporting

Needs improvements in writing and composing, literature, resutls

Experimental design

Methods described sufficient detail and information

Validity of the findings

data and experiments on it are valid

Additional comments

• Same citition in 0.2 Sequential recommendation (Rendle et al., 2010)
• Text and the refrences used in section 0.2 are irrelevant to this study, especially the 2nd paragraph as authors neither used Transformers in this study nor this study has any linked with Transformers. This should be removed.
• Typos on 158 line. 125, 171.,
• Use equation numbers
• All the equations are very difficult to understand. Terms in these equations are not well defined with respect to subject discuss in this manuscript particularly equations of section 1.4 and 1.5.
• Figure 1 should be devided into two Figures or subparts (Firstinteraction graph convolutional layer and second the other part on right side) and captioned them accordingly.
• How the 5th column (destiny) calculated? And what is its purpose…also provide its link to download and use for future research.
• Evaluation matrices need more explaination. How these metrics can be calculated and how these are useful for comparison of different mdoels.
• Table 2 should appear before the figure 2 as Table has been citing before the figure 2.

Cite this review as

---

## Round 0.2 · accepted · Accept

Thank the authors for their efforts to improve the work. Based on the comments of the reviewer and to the best of my knowledge, I believe the work has addressed the concerns proposed by the reviewers.

Reviewer 1 ·

Basic reporting

my concerns have been answered

Experimental design

no comment

Validity of the findings

no comment

Cite this review as